# Cryoprotective Effects of Carrageenan Oligosaccharides on Crayfish (*Procambarus clarkii*) during Superchilling

**DOI:** 10.3390/foods12112258

**Published:** 2023-06-03

**Authors:** E Liao, Yuxin Wu, Yang Pan, Ying Zhang, Peng Zhang, Jiwang Chen

**Affiliations:** 1College of Food Science and Engineering, Wuhan Polytechnic University, Wuhan 430023, China; leoneason@126.com (E.L.);; 2Hubei Key Laboratory for Processing and Transformation of Agricultural Products (Wuhan Polytechnic University), Wuhan 430023, China; 3National Research & Development Branch Center for Crayfish Processing (Qianjiang), Qianjiang 433100, China

**Keywords:** crayfish, cryoprotectant, carrageenan oligosaccharides, superchilling

## Abstract

Cryoprotectants are widely used to protect muscle tissue from ice crystal damage during the aquatic products freezing process, but traditional phosphate cryoprotectants may cause an imbalance in the calcium-to-phosphorus ratio for the human body. This study evaluated the effects of carrageenan oligosaccharides (CRGO) on quality deterioration and protein hydrolysis of crayfish (*Procambarus clarkii*) during superchilling. The physical-chemical analyses showed that CRGO treatments could significantly (*p* < 0.05) inhibit the increase of pH values, TVB-N, total viable counts, and thawing loss, and improve the water holding capacity and the proportion of immobilized water, which indicated that CRGO treatment effectively delayed the quality deterioration of crayfish. The myofibrillar protein structural results demonstrated that the increase of the disulfide bond, carbonyl content, S_0_-ANS, and the decrease of total sulfhydryl content were suppressed significantly (*p* < 0.05) in CRGO treatment groups. Furthermore, SDS-PAGE results showed that the band intensity of myosin heavy chain and actin in CRGO treatment groups were stronger than in the control. Overall, the application of CRGO to crayfish might maintain better quality and stable protein structure during the superchilling process, and CRGO has the potential to replace phosphate as a novel cryoprotectant for aquatic products.

## 1. Introduction

Crayfish (*Procambarus clarkii*) is one of the most popular aquatic products, with its output having shown an increasing trend in the past couple of years. In 2021, the annual capacity has been up to 2.63 million tons and the output value has reached 422 billion yuan. As important economic fishery resource, crayfish has high contents of moisture, amino acids, unsaturated fatty acids and superior protein (17.70% of wet basis). However, crayfish is prone to spoilage due to its high moisture content and the action of endogenous enzymes and microorganisms. The high levels of autolytic enzymes could lead to severe protein structure deterioration and give rise to a degradation in food quality. Therefore, to delay crayfish spoilage and physico-chemical changes, exploiting efficient preservative methods has become an area of interest.

Superchilling, also known as partial freezing, is potentially more attractive comparing with freezing, because fewer ice crystals formed resulting in a lower protein denaturation and less tissue damage [1]. Meanwhile, the speed of quality deterioration in superchilled products is slower than in conventionally refrigerated, and many researchers in the realm of food preservation are beginning to focus on superchilling technology [2]. Our previous research found that superchilling was effective in reducing trimethylamine (TMA), total volatile basic nitrogen (TVB-N), and biogenic amines of crayfish and also promoting the accumulation of inosine monophosphate (IMP) [3], which act as a flavor enhance [4]. However, compared with refrigeration, the temperature of superchilling was lower and below freezing, which could lead to the formation of ice crystals, bringing negative effects on crayfish muscle tissue.

Cryoprotectants were used to protect muscle tissue from damage during freezing. As a well-known cryoprotectant, phosphates play a widely beneficial role in water retention capacity, reducing thawing loss, and retarding protein hydrolysis. Etemadian et al. [5] found that sodium tripolyphosphate (STPP) as a cryoprotectant agent could significantly (*p* < 0.05) delay the damage of fish protein structure during the freezing process of Rutilus frisii kutum fillets. Sodium trimetaphosphate (STMP) also had a positive protective effect on frozen whiteleg shrimp [6]. Nevertheless, too much phosphorus in the human body would lead to an imbalance of calcium and phosphorus ratio, affect calcium absorption, and cause diseases in some people with special constitutions [7]. Consequently, the number of research into creating innovative cryoprotectants to maintain premium quality was growing rapidly. Cryoprotective saccharides are widely accepted as cryoprotectant additives [8], including chitosan nanoparticles [9], trehalose and alginate oligosaccharides [8], konjac glucomannan [10], and xylooligosaccharides [11]. Sun et al. [12] reported that carrageenan oligosaccharides (CRGO) possess high antioxidant activity and great potential for biomedical and physiological applications. Moreover, the CRGO treatment had an obvious protective effect on the muscle structure of frozen shrimp mud [13]. Zhang et al. [14] reported that CRGO, a carbohydrate with high biosafety, exhibited a certain degree of water-holding and tissue protection capacity for peeled shrimp (*Litopenaeus vannamei*) during long-term frozen storage. Nevertheless, there are few pertinent research papers reported concerning the application of oligosaccharides in crayfish during the superchilling process.

Therefore, the objective of this study was to observe the effect of CRGO on the quality of crayfish during superchilling by tracking changes in quality parameters and myofibrillar parameters, including pH, TVB-N, total viable count (TVC), thawing loss, water holding capacity (WHC), water migration, total sulfhydryl (T-SH), disulfide bond, carbonyl and surface hydrophobicity (S_0_-ANS).

## 2. Materials and Methods

### 2.1. Crayfish Samples and Treatments

Fresh crayfishes (*Procambarus clarkii*) weighing 25.0–30.0 g and 12.5–14.0 cm in length were obtained from a local supermarket in Wuhan of Hubei Province, China. Approximately 10 kg of live crayfish were packed in a foam box with trash ice and then transported to the laboratory within 30 min. Afterwards, the crayfish were placed in ice for cold shock (no more than 3 h) after being ultrasonically treated with tap water at 2.5 kHz for 30 min. When the heads, glands, and shells were removed, the muscles were placed into crushed ice instantly. Subsequently, the crayfish were submerged in prepared solutions at 0 to 4 °C and soaked for 30 min to allow the diffusion of the cryoprotective substances onto the surface of the crayfish with stirring every 10 min. The solutions were as follows: distilled water (group WNC), 1.0% (*w*/*v*) and 3.0% (*w*/*v*) STPP (purchased from Xiya Chemical Technology Co., LTD, Linyi, China), 1.0% (*w*/*v*) and 3.0% (*w*/*v*) CRGO (purity above 90%, molecular weight below 3000 Da, purchased from Qingdao BZ oligo Biotech Co., Ltd., Qingdao, China). After the soaking period, the crayfish were removed and drained for 1 min, packaged in polyethylene bags and stored at −3 ± 1 °C (a foam container containing 1.5% (*w*/*w*) ice–sodium chloride–water mixture at 4 °C). The refrigerated sample without treatment was blank control (group Control). Samples in each treatment were taken out randomly and three replications were analyzed on days 0, 3, 7, 14, and 21 of storage.

### 2.2. Determination of pH

The pH values were evaluated according to the method of Qin et al. [3]. Briefly, a high-speed disperser (Scientz, Ningbo, China) homogenized 5 g of crayfish muscles with 50 mL ultrapure water at 8000 r/min for 1 min, and a digital pH meter (Mettler-Toledo, Shanghai, China) was used to determine the pH value.

### 2.3. Determination of TVB-N

The TVB-N (mg N/100 g crayfish muscles) contents were determined according to Huang et al. [15] with minor modifications. Five grams of minced muscles were homogenized with 25 mL 20 g/L trichloroacetic acid at 8000 r/min for 1 min and stood for 1 h in the refrigerator. Afterward, fast filter paper with 11 cm was used to filter the homogenate. A Kjeldahl-type tube was loaded with 10 mL of the filtrate, 10 mL distilled water, and 5 mL 20 g/L MgO suspension for 5 min distillation. The distillate was collected using a flask with boric acid (10 mL, 20 g/L) and mixture indicators, and the solution in the flask was titrated with 0.01 M hydrochloric acid standard titrant.

### 2.4. Microbiological Analysis

The total viable count (TVC) was measured by the method of Yu et al. [16]. Samples of muscle (5 g) were mixed with 45 mL sterile saline water (0.85% NaCl) using a stomacher bag for 2 min. Then, the homogenates were serially diluted in sterile saline water at a ratio of 1:10. To determine TVC, 1 mL of the dilution was inoculated on PCA agar and incubated at 3 °C for 48 h. All counts were expressed as log_10_ CFU/g.

### 2.5. Determination of Thawing Loss

The thawing loss was assessed by Ma et al. [17] with many alterations. The samples under different storage periods were taken out, weighed as M_1_, thawed at 25 °C for 1 h, and drained with kitchen paper. The mass of the samples was accurately weighed as M_2_. The thawing loss of the superchilled crayfish was calculated immediately as follows: X = (M_1_ − M_2_)/M_1_ × 100.

### 2.6. Determination of Water Holding Capacity

The water holding capacity (WHC) of crayfish was estimated as reported by the method of Gao et al. [18] with a modification. Summarily, 2 g sample of crayfish muscle with one layer of 11 cm fast filter paper was placed in a 10 mL centrifuge tube and centrifuged at 5000 rpm for 10 min at 4 °C. The ratio of sample weight after centrifugation to sample weight before centrifugation is the WHC.

### 2.7. Low Field ^1^H Nuclear Magnetic Resonance (LF ^1^H NMR) Spectroscopy

To identify water movement and redistribution in crayfish, a 23 MHz NMR spectroscopy (NMI20-040V-I, Niumag, Shanghai, China) was used [19]. The Carr–Purcell–Meiboom–Gill pulse sequence was used to measure the spin-spin relaxation time (T_2_), and the following settings were used: PRG = 2, RFD = 0. 002 ms, TE (ms) = 0.150, Tw = 1500 ms, NECH = 12,000, NS = 8. T_21_, T_22,_ and T_23_ represented the T_2_ of bound water, immobilized water, and free water, respectively.

### 2.8. Extraction of Myofibrillar Protein

Myofibrillar protein was extracted from crayfish as described by Jiang et al. [20] with some modifications. Briefly, a high-speed disperser (Scientz, Ningbo, China) was used to homogenize 3.0 g of each minced crayfish muscle for 60 s at 8000 r/min in 30 mL of ice-cold buffer A (pH 7.5, 20 mmol/L Tris-maleate containing 0.1 mol/L KCl). The homogenate was centrifuged at 6000× *g* for 20 min at 4 °C. The supernatant was discarded, and the precipitate was used to repeat the above procedures in the ice-cold buffer B (pH 7.0, 20 mmol/L Tris-maleate containing 0.6 mol/L KCl) to extract again. The myofibrillar protein solution, which was from the second supernatant, was kept at a temperature between 0 °C and 4 °C and then utilized for the following analyses. Using bovine serum albumin as a reference, the Biuret method was used to quantify the content of myofibrillar protein in the supernatant.

### 2.9. Determination of Total Sulfhydryl Content

The method of 5, 5′-dithiobis-(2-nitrobenzoic acid) (DTNB) as described by Zhang et al. [14] was slightly modified to quantify the total sulfhydryl (T-SH) concentration. Briefly, T-SH buffer A (9 mL) containing 8 mol/L urea, 2% (*w*/*v*) sodium dodecyl sulfate (SDS), 10 mmol/L EDTA, and 0.2 mol/L Tris-HCl (pH 7.0) was combined with a myofibrillar protein solution (1 mL, roughly 0.5 to 2 mg protein/mL). The mixture was vortexed for 1 min at 201× *g* using a vortex shaker (Xinbao, China), then allowed to stand for 30 min at ambient temperature. Then, 4 mL of the mixture was added to 1 mL of T-SH buffer B (2.5 mmol/L DTNB and 0.2 mol/L Tris-HCl, pH 8.0) and incubated at 40 °C for 25 min. The reaction was carried out the same way for the blank, except the ice-cold buffer B was used to substitute the myofibrillar protein. Using an ultraviolet-visible spectrophotometer (UV-5100, METASH, Shanghai, China), absorbance was measured at 412 nm. These steps were used to calculate T-SH content: total SH content (mol/10^5^ g) = A × D/ (ξ × C), where A represents the absorbance at 412 nm, C represents the myofibrillar protein concentration (mg/mL), ξ represents the molar extinction coefficient (13,600 mol^−1^·cm^−1^·L), and D represents the dilution ratio.

### 2.10. Determination of Disulfide Bond Content

Disulfide bond content was determined using the 2-nitro-5-thiosulphobenzoate (NTSB) as described by Riebroy et al. [21]. To 1 mL of protein sample (0.5 to 2 mg /mL), 4 mL of NTSB assay solution (freshly prepared, pH 9.5) was appended. The hybrid was cultivated in the dark at room temperature for 30 min, and then the absorbance was estimated at 412 nm. The disulfide bond content was calculated using a molar extinction coefficient (13,600 mol^−1^·cm^−1^·L) and, in this way, the disulfide bond content (mol/10^5^ g) = A × D/(ξ × C). The letters represent the same meaning as the total sulfhydryl group.

### 2.11. Determination of Carbonyl Content

According to Kong et al. [22] with slight modifications, the samples were combined with 2, 4-dinitrophenylhydrazine (DNPH) to create protein hydrazones, and the carbonyl concentration was then determined by spectrophotometry. The mixture of 1 mL of myofibrillar protein solution and 1 mL of DNPH (10 mmol/L) in 2 mol/L of hydrochloric acid was kept out of the light for 1 h, and then 20% trichloroacetic acid (1 mL) was poured into the mixture. After centrifugation (5000 rpm, 10 min), the sediments were rinsed thrice with 1 mL of ethanol: ethyl acetate (1:1, *v*/*v*) to remove the DNPH, and then taken a water bath at 37 °C for 30 min to dissolve the precipitation after adding 4 mL guanidine hydrochloride. The absorbance of the supernatant was determined at 370 nm after centrifugation (10,000 rpm, 5 min) with HCl (2 mol/L) as the control. The carbonyl content was expressed as follows: carbonyl content (mol/10^5^ g) = A/(ε × C), where A is the absorbance at 370 nm, C is the concentration of myofibrillar protein (mg/mL), and ε is the absorption coefficient of 22,000 mol^−1^·cm^−1^·L for protein hydrazones.

### 2.12. Determination of Surface Hydrophobicity

As a sensitive indication of minute changes in the physical and chemical states of myofibrillar proteins, surface hydrophobicity alterations can be utilized to track conformational changes in protein structure [23]. According to Zhang et al. [24] with minor changes, 1-anilinonaphthalene-8-sulfonic acid (ANS) was utilized as the fluorescent probe, and the surface hydrophobicity of myofibrillar protein under various periods of storage was investigated. The myofibrillar protein solution was diluted with the ice-cold buffer B (pH 7.0) to 0.3 mg/mL and then attenuated to obtain a series of protein concentrations, from 0 to 1 mg/mL. To 4 mL of each protein solution, 20 µL of 8 mM ANS (keep in a dark place) in 0.1 M phosphate buffer (pH 7.0) was added and mixed thoroughly. The relative fluorescence intensity of protein-ANS compounds was immediately estimated using a luminescence spectrophotometer (F-4600 model, Hitachi, Tokyo, Japan) at excitation and emission wavelengths of 385 nm and 470 nm, respectively, and a 2.5 nm width for both the excitation and emission slits. Using linear regression analysis, the initial slopes of plots of relative fluorescence intensity versus protein concentration were used to determine the surface hydrophobicity of myofibrillar protein, the S_0_-ANS was the initial slope.

### 2.13. Sodium Dodecyl Sulfate-Polyacrylamide Gel Electrophoresis (SDS-PAGE) Analysis

SDS-PAGE was carried out based on the method of Qi et al. [25] with 12% separating gel and 5% stacking gel. Myofibrillar protein solution (1 mg/mL) was blended with a quarter volume of loading buffer (Beyotime, Shanghai, China). After incubation for 5 min at 95 °C, 10 μL of each sample was loaded in the sample well, and the mobility of the protein bands was calibrated with standards of molecular weight markers. Electrophoresis was initially run at a voltage of 70 V (PowerPac TM HC, Bio-Rad, Singapore) for 30 min, then the voltage was increased to 120 V for 60 min. The gel was stained with 0.1% Coomassie brilliant blue R-250 (Solarbio, Beijing, China) in 45% ethanol (*v*/*v*) and 10% acetic acid (*v*/*v*) for 30 min and destained with the solution (10% methanol, 10% acetic acid). The photo was obtained using a gel imaging and analysis system (Universal Hood III, Bio-Rad, Wuhan, China).

### 2.14. Statistical Analysis

Each experiment was carried out three times. The means ± standard deviation of the data was provided after analysis with the SPSS 19.0 software. The one-way ANOVA and Duncan’s multiple-range tests were used to determine the significance of differences. Significant was defined to be *p* < 0.05.

## 3. Results and Discussion

### 3.1. The pH, TVB-N, and TVC Analysis

The changes in pH values of samples during storage are shown in Figure 1a. Significant changes (*p* < 0.05) in pH were observed during 21 days of storage. The initial pH of fresh crayfish was 6.58 ± 0.02, similar to the result reported by Nirmal et al. [26] for Pacific white shrimp. The higher initial pH was observed in STPP pretreatments, which might be attributed to the alkali properties of phosphates. The pH values decreased slightly at first (≤3 days) and then increased gradually in Control and WNC, and these increases were delayed in all superchilled samples to days 7–14. The incipient decrease might be related to the accumulation of lactic acid during anaerobic glycolysis and the breakdown of ATP, and subsequent increases could attribute to the generation of alkaline metabolites (biogenic amines, trimethylamine, etc.) by the metabolism of microorganisms [3]. After 21 days of storage, the pH values in CRGO and STPP are much lower than those in Control and WNC, which indicated that CRGO and STPP could slow microbial reproduction and protein enzyme activity.

TVB-N such as dimethylamine, TMA, and ammonia are produced during the decay of protein foods [27]. Hence, TVB-N could be an important indicator to reflect the corruption degree of crayfish. As shown in Figure 1b, there are continuous increases of TVB-N values for all groups throughout the storage, but the increase rates are varied among different treatments. The initial value of TVB-N in fresh crayfish was 4.63 mg/100 g, which was close to the value of Wuchang bream (6.3 mg/100 g) [15], which indicates the freshness of samples was acceptable. In all superchilled groups, the TVB-N values ranged from 4.63 mg/100 g at day 0 to 12 mg/100 g at 21 d, which are not exceeded the acceptable limit of China (20 mg/100 g) [15], and no significant differences (*p* > 0.05) were observed among those groups. However, significantly higher (*p* < 0.05) TVB-N content was found in the control, which showed that superchilling could efficaciously impede the rapid accumulation of TVB-N by preventing protein degradation and decarboxylation of amino acids.

The variations of TVC In all samples during storage were investigated to judge the spoilage degree of crayfish. As shown in Figure 1c, the TVC increased steadily in all groups during whole storage. The microbial growth rate was faster in the crayfish stored at 4 °C than in superchilled samples, in which TVC increased from an initial level of 2.26 log_10_ CFU/g to 8.50 log_10_ CFU/g at the end of the storage, exceeding the highest limit of 7 log_10_ CFU/g [28]. The similar results about initial TVC levels have been found by Nirmal et al. [29] in *Litopenaeus vannamei* (2.5 log_10_ CFU/g) and by Li et al. [30] for Pacific white shrimp (2.42 log_10_ CFU/g). In all superchilled samples, the TVC values were significant difference (*p* < 0.05) at the end of the storage, the values were 4.50, 4.33, 4.27, 4.32, and 4.15 log_10_ CFU/g, respectively. The microbial levels in CRGO and STPP were lower than those in WNC, which could be attributed to the anti-bacterial activity of phosphates [31] and CRGO [32]. Furthermore, the lowest TVC value was observed in 3% CRGO, which demonstrated that the anti-bacterial activity of CRGO was positively correlated with its concentration.

### 3.2. WHC and Thawing Loss Analysis

The decrease of WHC in the muscle of aquatic products is often described as the effect of structural changes in muscle during storage, which can result in myofibril network contraction, myosin protein deformation, and an increased extracellular space [33]. As shown in Figure 2a, the WHC of all groups decreased significantly (*p* < 0.05) with the extension of storage time from the initial value of 84.58% to 61.11–71.44% at the end of storage. There was no significant difference (*p* > 0.05) among the groups at the beginning of storage (≤7 days), but group WNC was downgraded to 61.11% on day 21, which was significantly (*p* < 0.05) lower than other groups. The results indicated that STPP and CRGO treatments could improve the WHC of crayfish during superchilling. In addition, the water retention of 3% cryoprotectant treatment was significantly (*p* < 0.05) higher than 1% treatment, and the effect was a positive correlation with the concentration of cryoprotectant and no significant difference (*p* > 0.05) was observed between 3% STPP and 3% CRGO.

The superchilling temperature is below freezing, causing a partial freeze of samples, so the changes in the thawing loss were investigated in all superchilled samples, and the results are shown in Figure 2b. During the whole storage process, the thawing loss of each group increased gradually, and the highest value (24.18%) was observed in WNC on the 21st day. Smaller water loss could be found in CRGO and STPP, ranging from 11.50% to 19.05%, possibly demonstrating that STPP and CRGO treatments could inhibit the damage of myofibrils tissue caused by ice crystals [34]. In conclusion, there is a negative correlation between thawing loss and the WHC.

### 3.3. Water Mobility and Redistribution

To reveal the effect of different cryoprotectants on the water migration of crayfish, LF ^1^H NMR was used to analyze the relaxation time (T_2_) of samples over 21 days of storage, and the results are shown in Figure 3a–c. There are three peaks, which, respectively represent bound water (T_21_, 0–10 ms), immobilized water (T_22_, 60–200 ms, water stabilized in the gel network), and free water (T_23_, 600–1000 ms). Among them, immobilized water was the main water component in crayfish. With the prolonged storage time, the contents of bound water in all groups basically remained stable, but immobilized water decreased significantly (*p* < 0.05). After 21 days of storage, the proportion of non-free water in CRGO was significantly (*p* < 0.05) higher than that in WNC, which was similar to that in STPP, indicating that CRGO has a positive effect in water fixation similar to STPP, and the degree of fixation was positively correlated with the concentration levels.

The peak area ratio of P_21_, P_22,_ and P_23_ over 21 days of superchilling are shown in Figure 3d. During the storage, the proportion of non-free water in six groups decreased slightly, from 99.54% ± 0.36% to 98.76%, 98.28%, 98.87%, 99.57%, 99.46%, and 99.82%, respectively. After 21 days of storage, the P_21_ of Control decreased to 5.167%, which was significant (*p* < 0.05) lower than that of the STPP and CRGO. The results indicated that STPP and CRGO treatment attenuates the migration of bound water during storage. Meanwhile, the proportion of immobilized water in WNC was significantly (*p* < 0.05) lower than that in 1% STPP and 1% CRGO. However, the high proportion of free water contained in 3% CRGO may be due to the cryoprotectant could protect cell structure and inhibit the loss of free water.

### 3.4. Total Sulfhydryl and Disulfide Bond Analysis

Myofibrillar protein comprises a large number of sulfhydryl groups [35]. The changes in sulfhydryl contents are often used as the index of protein degradation. Figure 4a presents the changes in total sulfhydryl (T-SH) contents, and the overall trends are downward over 21 days. The original average value of T-SH was 4.61 ± 0.20 mol/10^5^ g protein, similar to the value in *Litopenaeus vannamei* reported by Zhang et al. [14], indicating a lower degree of protein oxidation. Within 21 days of storage, the T-SH concentration in Control and WNC declined severely, while that in CRGO and STPP decreased tardily. At the end of the storage, the T-SH concentrations in 1% CRGO, 3% CRGO, 1% STPP, and 3% STPP were reduced by 26.43%, 23.48%, 27.05%, and 25.51%, respectively, and significantly (*p* < 0.05) higher than Control (28.09%) and WNC (27.42%). In conclusion, the decrease of the T-SH could be attributed to the formation of disulfide bonds either within polypeptides or between polypeptides, or its own degradation reactions [36].

The changes in disulfide bond contents are depicted in Figure 4b, and the trends were negatively correlated with that of T-SH contents. The initial average value of the disulfide bond was 2.76 ± 1.20 mol/10^5^ g protein. Generally, the disulfide bond contents increased significantly (*p* < 0.05) with the prolongation of storage time. The values on the 21st day reached to range from 4.39 to 5.50 mol/10^5^ g protein, and the highest value was observed in Control, suggesting a higher degree of protein oxidation. Meanwhile, the disulfide bond contents in 3% STPP and 3% CRGO reached 4.39 and 4.60, respectively, which could demonstrate that the cryoprotectant treatments had an obvious effect on inhibiting the accumulation of disulfide bond, and higher concentration was more efficient in suppression of sulfhydryl oxidation. In addition, there was no evident difference (*p* > 0.05) between 3% CRGO and 3% STPP.

### 3.5. Carbonyl Content and Surface Hydrophobicity Analysis

The carbonyl groups may serve as a significant reactive group in forming peptide bonds and represent the level of protein oxidation [37]. Therefore, the degree of protein oxidation in crayfish stored in superchilling with various treatments was assessed via its carbonyl content. As shown in Figure 5a, the carbonyl contents of all groups augment significantly (*p* < 0.05) during the whole storage process and increased from 0.15 mol/10^5^ g to 0.72, 0.76, 0.66, 0.57, 0.65, and 0.59 mol/10^5^ g protein, respectively. The rates of carbonyl contents growth in all treatments with cryoprotectant were slower than that in the Control and WNC. Among them, 3% CRGO and 3% STPP have lower carbonyl contents at the end of the storage. The above results show that the CRGO and STPP treatments obstructed the carbonyl formation effectively, which demonstrated their effectiveness in the suppression of protein denaturation, and the degree of suppression is positively correlated with concentration.

Surface hydrophobicity (S_0_-ANS) can indicate how exposed hydrophobic amino acids are to the environment in myofibrillar protein. Hydrophobic amino acids are present in the protein folding core by nature, but structural modifications could make them accessible to myofibrillar protein surfaces [24]. As shown in Figure 5b, the surface hydrophobicity increased gradually in all groups with the extension of storage, from the initial value 594.35 ± 2.50 to 730.55, 709.03, 693.69, 665.64, 699.76, 670.64, respectively, proving that superchilling could retard the exposure of hydrophobic groups. The S_0_-ANS values in CRGO showed a slower increase compared to WNC. A similar result was reported by Zhang et al. [38], and they found that kappa-carrageenan oligosaccharides could delay the oxidation of protein and lipids in mackerel (*Scomber japonicus*) during frozen storage. Furthermore, the inhibition effect of 3% CRGO on S_0_-ANS values was more obvious. Nevertheless, there was no significant (*p* > 0.05) difference between 3% CRGO and 3% STPP, which explained that CRGO has the potential as a cryoprotectant to replace the use of conventional phosphate.

### 3.6. SDS-PAGE

Changes in myofibrillar protein in all groups are shown in Figure 6. Several bands from the samples were visible in the SDS-PAGE pattern, and the two bands with the highest optical densities were myosin heavy chain (MHC, 220 kDa) and actin (43 kDa) [39]. Generally, the band intensity of MHC and actin in 3% STPP and 3% CRGO were stronger, and no evident difference was observed among the other groups at the end of storage. In addition, the bands with molecular weights of around 16 kDa are lighter in the Control and WNC, and apparent breakdowns were observed around 80 kDa bands of myofibrillar proteins in the Control. In comparison, no obvious change of bands was observed in CRGO treatments, which further suggested that these treatments efficiently suppressed the hydrolysis of myofibrillar protein. Hence, it can potentially replace phosphate as a new natural cryoprotectant for aquatic products.

## 4. Conclusions

In this study, the effects of CRGO on quality deterioration and protein hydrolysis of crayfish during the superchilling process were investigated by analyzing the quality-related indexes (pH, TVB-N, TVC, WHC, thawing loss, and water status) and protein structure-related indexes (T-SH, disulfide bond, carbonyl content, and S_0_-ANS). The increase of pH values, TVB-N, TVC, and thawing loss were significantly (*p* < 0.05) inhibited in CRGO treatment groups. The WHC and the proportion of immobilized water in CRGO treatment groups were significantly (*p* < 0.05) higher than in control throughout the superchilling process, which indicated that CRGO treatment could effectively inhibit the quality deterioration of crayfish during superchilling. In addition, the decrease of T-SH contents and increase of disulfide bond contents were significantly (*p* < 0.05) suppressed in CRGO treatment groups. The lowest carbonyl content (0.57 mol/10^5^ g protein) and S_0_-ANS (665.64) were observed in 3% CRGO at the end of the storage, which showed that the protein of the sample treated with 3% CRGO had the lowest oxidation level after 21 days superchilling. Furthermore, SDS-PAGE results revealed that the degradation of MHC and actin was lower in CRGO treatment groups than in the control. Overall, the application of 3% CRGO has an obvious effect on retaining moisture and retarding hydrolysis of myofibrillar protein in crayfish during superchilling, which indicated that CRGO has the potential as a novel cryoprotectant to replace phosphate in aquatic products.

## Figures and Tables

**Figure 1 foods-12-02258-f001:**
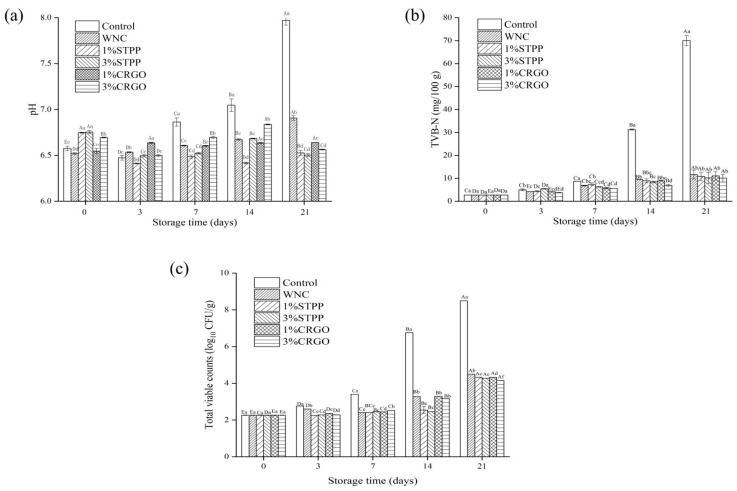
The pH (**a**), total volatile base nitrogen (TVB-N) (**b**), and total viable counts (TVC) (**c**) of crayfish for 21 days storage. Different capital letters within the same treatment indicate significant differences (*p* < 0.05). The different lowercase letters within the same storage time indicate significant differences (*p* < 0.05). Control, refrigerated samples, WNC, and superchilled samples were treated with distilled water, 1%STPP, and 3%STPP, superchilled samples were treated with 1% and 3% sodium tripolyphosphate, 1%CRGO, and 3%CRGO, superchilled samples were treated with 1% and 3% carrageenan oligosaccharides.

**Figure 2 foods-12-02258-f002:**
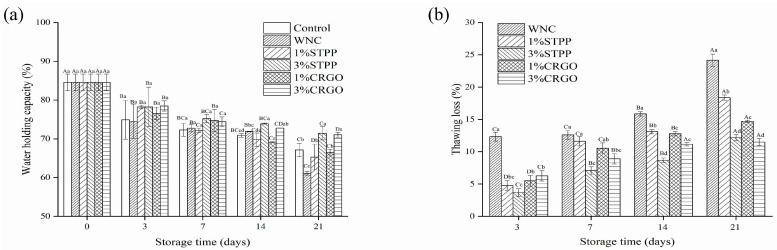
The water holding capacity (WHC) (**a**) and thawing loss (**b**) of crayfish over 21 days. Different capital letters within the same treatment indicate significant differences (*p* < 0.05). The different lowercase letters within the same storage time indicate significant differences (*p* < 0.05). Control, refrigerated samples, WNC, and superchilled samples were treated with distilled water, 1% STPP, and 3% STPP, superchilled samples were treated with 1% and 3% sodium tripolyphosphate, 1% CRGO, and 3% CRGO, superchilled samples were treated with 1% and 3% carrageenan oligosaccharides.

**Figure 3 foods-12-02258-f003:**
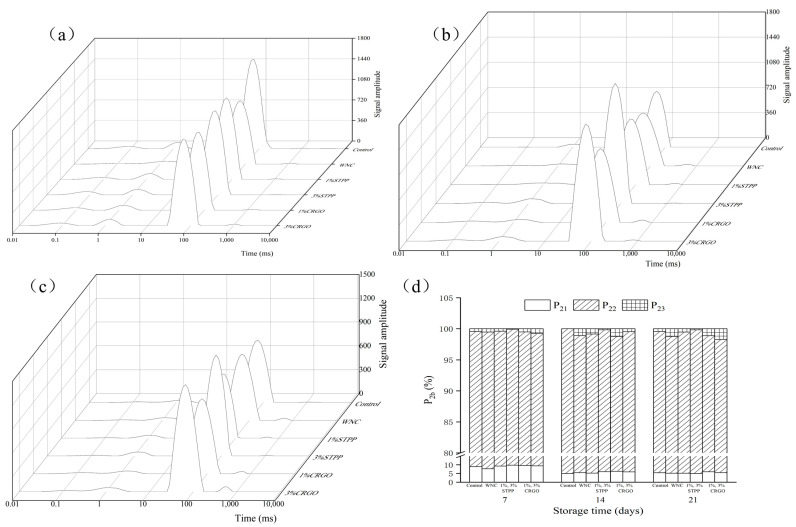
LF ^1^H NMR relaxation time (T_2b_) (**a**–**c**) and peak ratio (P_2b_) (**d**) of crayfish during 21 days. The (**a**–**c**) showed the relaxation time (T_2b_) of crayfish stored for 7, 14, and 21 days, respectively. Different capital letters within the same treatment indicate significant differences (*p* < 0.05). The different lowercase letters within the same storage time indicate significant differences (*p* < 0.05). Control, refrigerated samples, WNC, and superchilled samples were treated with distilled water, 1% STPP, and 3% STPP, superchilled samples were treated with 1% and 3% sodium tripolyphosphate, 1% CRGO, and 3% CRGO, superchilled samples were treated with 1% and 3% carrageenan oligosaccharides.

**Figure 4 foods-12-02258-f004:**
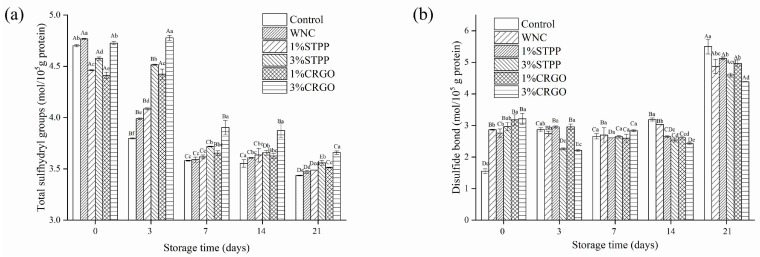
The total sulfhydryl content (**a**) and disulfide bond content (**b**) of myofibrillar protein of crayfish muscle over 21 days. Different capital letters within the same treatment indicate the significant differences (*p* < 0.05). The different lowercase letters within the same storage time indicate significant differences (*p* < 0.05). Control, refrigerated samples, WNC, superchilled samples were treated with distilled water, 1% STPP and 3% STPP, superchilled samples were treated with 1% and 3% sodium tripolyphosphate, 1% CRGO and 3% CRGO, superchilled samples were treated with 1% and 3% carrageenan oligosaccharides.

**Figure 5 foods-12-02258-f005:**
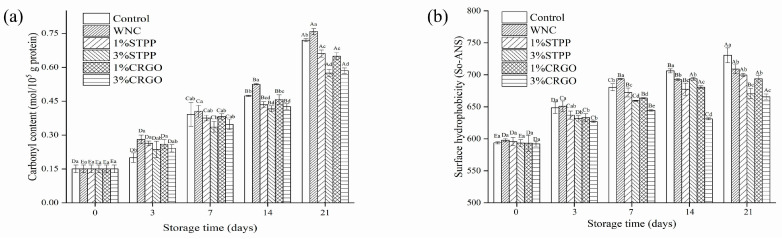
The carbonyl content (**a**) and surface hydrophobicity (S_0_-ANS) (**b**) of myofibrillar protein of crayfish muscle over 21 days. Different capital letters within the same treatment indicate significant differences (*p* < 0.05). The different lowercase letters within the same storage time indicate significant differences (*p* < 0.05). Control, refrigerated samples, WNC, and superchilled samples were treated with distilled water, 1% STPP, and 3% STPP, superchilled samples were treated with 1% and 3% sodium tripolyphosphate, 1% CRGO, and 3% CRGO, superchilled samples were treated with 1% and 3% carrageenan oligosaccharides.

**Figure 6 foods-12-02258-f006:**
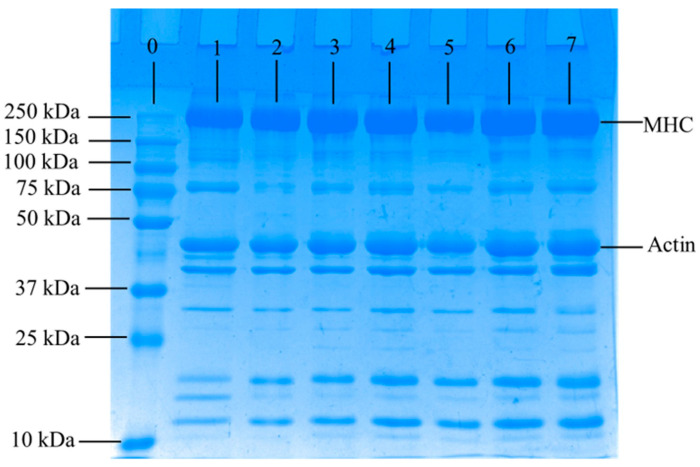
Changes in SDS-PAGE of myofibrillar protein of crayfish muscle over 21 days (Lane 0 marker, Lane 1 fresh sample, Lane 2 Control 21 days, Lane 3 WNC 21 days, Lane 4 1% CRGO 21 days, Lane 5 1%STPP 21 days, Lane 6 3% CRGO 21 days, Lane 7 3% STPP 21 days). Control, refrigerated samples, WNC, and superchilled samples were treated with distilled water, 1% STPP, and 3% STPP, superchilled samples were treated with 1% and 3% sodium tripolyphosphate, 1% CRGO, and 3% CRGO, superchilled samples were treated with 1% and 3% carrageenan oligosaccharides.

## Data Availability

The research data supporting this study can be found by contacting the corresponding author.

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
