# Peer review of "Cryoprotective Effects of Carrageenan Oligosaccharides on Crayfish (Procambarus clarkii) during Superchilling"

_foods, 2023, doi:10.3390/foods12112258_

Round 1

Reviewer 1 Report

1.    Why was carrageenan evaluated as a cryoprotectant?, are there doubts about its use in food products related to appearance of inflammation, possible ulcers generation, among others? This should be included in the Introduction section.

2.   The amount of carrageenan in the best treatment, meets the acceptable daily intake of ~75 mg/Kg.day?.

3.      More detail of carrageenan oligosaccharides, purity?

4.      Line 72: to eliminate parenthesis  migration)

5.      Line 73: to eliminate parenthesis … S0-ANS))  

6.      How much sample was used for each treatment? Indicate repetitions per treatment? How was the sampling done?

7.      Line 96, change Metler-Tolido to Mettler-Toledo

8.      In case it’s not indicated in analysis methods section, declare the units in which the results are expressed.

9.      Improve the quality of the figures, all of them are not clear.

10.  Line 232: to indicate examples of alkaline metabolites.

11.  Figure 1. Explain why distilled water treatment present results close to STTP and/or CRGO.

12.  Figure 2b, missing control data

13.  Something curious that can be observed from the data is that the WNC treatment results in some cases close to the use of STTP and/or CRGO, why could be happening? Distilled water would also be impacting the treatments where STTP or CRGO were used?.

Author Response

Dear Editors and Reviewers,

We are quite grateful to both your help and that of the reviewers concerning improvement to our manuscript entitled "Cryoprotective Effects of Carrageenan Oligosaccharides on Crayfish (Procambarus clarkii) during Superchilling" (foods-2393185). Those comments are all valuable and very helpful for revising and improving our manuscript, as well as the important guiding significance to our further research and writing. We have carefully addressed the comments raised by the reviewers, and the amendments are highlighted in yellow in the revised manuscript. The main corrections in the paper and the details of response to the reviewers' comments are shown below. Thanks for all the help and look forward to hearing from you soon.

With best wishes,

Sincerely yours,

Jiwang Chen

Following are our responses about editors' and reviewers' comments to our manuscript foods-2393185:

Reviewer 1:

Why was carrageenan evaluated as a cryoprotectant?, are there doubts about its use in food products related to appearance of inflammation, possible ulcers generation, among others? This should be included in the Introduction section.

Response

Thank you for your suggestion. The carrageenan oligosaccharides were evaluated as cryoprotectants because of it has been reported to exhibit a certain degree of water-holding and tissue protection capacity in frozen aquatic products. Besides, it was considered to have a strong biosafety, which would not cause inflammation, ulcers and other diseases. Relevant expressions and reference have been added in the introduction, lines 66-68.

The amount of carrageenan in the best treatment, meets the acceptable daily intake of ~75 mg/Kg.day?

Response

Thank you for your suggestion. In our treatments, 3% carrageenan oligosaccharides refer to the concentration of the solution in which the crayfish is soaked, the amount migrating to the crayfish during soaking process should be far below 3%. In addition, ~75 mg/Kg day is the acceptable daily intake for carrageenan, and the ADI of carrageenan oligosaccharides have not been reported. Of course, it is necessary to study the migration pattern of carrageenan oligosaccharides and the residues in crayfish, and we plan to clarify them in the subsequent experiments.

More detail of carrageenan oligosaccharides, purity?

Response

Thank you for your suggestion. More detail of carrageenan oligosaccharides, such as purity, molecular weight, etc. have been added in line 89.

Line 72: to eliminate parenthesis…migration)

Line 73: to eliminate parenthesis…S0-ANS))

Response

Thank you for your suggestion. The parenthesis have been eliminate, and the sentences have been modified slightly, lines 72-75.

How much sample was used for each treatment? Indicate repetitions per treatment? How was the sampling done?

Response

Thank you for your suggestion. The total amount of crayfish muscle weighed about 1200 g and were randomly divided into 6 groups of about 200 g each. Three replications of each treatment were carried out after random sampling. Relevant expressions have been added in the section of Materials and Methods, lines 93-95.

Line 96, change Metler-Tolido to Mettler-Toledo

Response

Thank you for your suggestion. “Metler-Tolido” has been edited as “Metler-Tolido” in line 99.

In case it’s not indicated in analysis methods section, declare the units in which the results are expressed

Response

Thank you for your suggestion. Surface hydrophobicity is indicated by the slope of the curve which is plotted with myogenic fibronectin as the horizontal coordinate and the fluorescence emission intensity at 470 nm as the vertical coordinate. Therefore, the surface hydrophobicity has no limited unit.

Improve the quality of the figures, all of them are not clear.

Response

Thank you for your suggestion. The quality of all figures have been improved to ensure a clear display.

Line 232: to indicate examples of alkaline metabolites.

Response

Thank you for your suggestion. The alkaline metabolites produced during spoilage of crayfish and other aquatic products mainly include biogenic amines, trimethylamine, and so on. The examples of alkaline metabolites have been indicated in line 236.

Figure 1. Explain why distilled water treatment present results close to STPP and/or CRGO.

Response

Thank you for your suggestion. Among the six groups designed for this experiment, group Control was stored in refrigerated condition and the other five groups were stored in superchilled condition. There was no significant difference (P > 0.05) in TVB-N values among the five treatments after 21 days under superchilling, probably due to the fact that superchilling significantly (P < 0.05) inhibited spoilage deterioration of crayfish. Meanwhile, the improve of carrageenan oligosaccharides and STPP on crayfish quality were more in terms of increasing water-holding capacity and reducing tissue damage. Therefore, the indicators subsequently analyzed in this study mainly focus on the water status and myofibrillar protein structure, such as total sulfhydryl contents, carbonyl contents, SDS-PAGE, and so on.

Figure 2b, missing control data

Response

Thank you for your suggestion. Figure 2b showed the results for the thawing loss rate, which could be measured in materials stored below freezing point, such as all treatments under superchilling in this study. However, Control was refer to the sample stored under refrigerated condition (4℃), and could not be measured this indicator. Therefore, in figure 2b, the WNC group was used as the control.

Something curious that can be observed from the data is that the WNC treatment results in some cases close to the use of STPP and/or CRGO, why could be happening? Distilled water would also be impacting the treatments where STPP or CRGO were used?

Response

Thank you for your suggestion. A large number of studies have shown that both STPP and carbohydrates could effectively delay the deterioration of protein structure during frozen storge, and the effects were positively correlated with the concentration. Some indicators such as disulfide bond contents and surface hydrophobicity showed no significant difference (P > 0.05) between the low concentration (1%) of cryoprotectant treatments and WNC at 21 d in superchilling, indicating that the cryoprotectant treatments at low concentration (1%) had a limited effect on the maintenance of crayfish quality under superchilling. Therefore, higher concentration of cryoprotectant (3%) was also applied in this study, and significant differences (P < 0.05) were observed between treatments with high concentration of cryoprotectant (3%) and WNC.

Reviewer 2 Report

This paper "Cryoprotective Effects of Carrageenan Oligosaccharides on Crayfish (Procambarus clarkii) during Superchilling" deals with the replacement of phosphate cryoprotectants with Carrageenan Oligosaccharides. 

The article is well written, however the quality of figures are not up to the mark. These figures are quite blurred and not readable. 

Results have been explained in a good manner, however discussion part is quite poor. The author must interpret the results and give Scientific evidence behind effect of CRGO on different parameters. 

Author Response

Dear Editors and Reviewers, We are quite grateful to both your help and that of the reviewers concerning improvement to our manuscript entitled "Cryoprotective Effects of Carrageenan Oligosaccharides on Crayfish (Procambarus clarkii) during Superchilling" (foods-2393185). Those comments are all valuable and very helpful for revising and improving our manuscript, as well as the important guiding significance to our further research and writing. We have carefully addressed the comments raised by the reviewers, and the amendments are highlighted in yellow in the revised manuscript. The main corrections in the paper and the details of response to the reviewers' comments are shown below. Thanks for all the help and look forward to hearing from you soon. With best wishes, Sincerely yours, Jiwang Chen   Following are our responses about editors' and reviewers' comments to our manuscript foods-2393185: Reviewer 2: This paper "Cryoprotective Effects of Carrageenan Oligosaccharides on Crayfish (Procambarus clarkii) during Superchilling" deals with the replacement of phosphate cryoprotectants with Carrageenan Oligosaccharides. The article is well written, however the quality of figures are not up to the mark. These figures are quite blurred and not readable. Results have been explained in a good manner, however discussion part is quite poor. The author must interpret the results and give Scientific evidence behind effect of CRGO on different parameters. Response: Thank you for your suggestion. The quality of all figures have been improved to ensure a clear display. Meanwhile, we have revised the discussion section as much as possible, and some scientific evidences were added to support the correlation between CRGO and related indicators. The amendments are highlighted in yellow in the revised manuscript.

Reviewer 3 Report

The objective of the paper is clear and of significant interest for food industry and environmental protection. The experimental data are obtained trough a good protocols and scientific interpretation is well conducted.

Minor edidint required 

Author Response

Dear Editors and Reviewers,

We are quite grateful to both your help and that of the reviewers concerning improvement to our manuscript entitled "Cryoprotective Effects of Carrageenan Oligosaccharides on Crayfish (Procambarus clarkii) during Superchilling" (foods-2393185). Those comments are all valuable and very helpful for revising and improving our manuscript, as well as the important guiding significance to our further research and writing. We have carefully addressed the comments raised by the reviewers, and the amendments are highlighted in yellow in the revised manuscript. The main corrections in the paper and the details of response to the reviewers' comments are shown below. Thanks for all the help and look forward to hearing from you soon. 

With best wishes,

Sincerely yours,

Jiwang Chen

Following are our responses about editors' and reviewers' comments to our manuscript foods-2393185:

Reviewer 3:

The objective of the paper is clear and of significant interest for food industry and environmental protection. The experimental data are obtained trough a good protocols and scientific interpretation is well conducted.

Response

Thank you for your positive comments on this study. We have double-checked the full manuscript and adjusted some phrases to make it more accord with the requirements of academic standardization, and the amendments are highlighted in yellow in the revised manuscript.

Round 2

Reviewer 1 Report

Accept

Author Response

Thank you for your comment.